# Trade-Off Between Enzymatic Antioxidant Defense and Accumulation of Organic Metabolite Affects Salt Tolerance of White Clover Associated with Redox, Water, and Metabolic Homeostases

**DOI:** 10.3390/plants14020145

**Published:** 2025-01-07

**Authors:** Min Zhou, Yuting Wu, Yuchen Yang, Yan Yuan, Junnan Lin, Long Lin, Zhou Li

**Affiliations:** Department of Turf Science and Engineering, College of Grassland Science and Technology, Sichuan Agricultural University, Chengdu 611130, China; 2021202098@stu.sicau.edu.cn (M.Z.); wuyuting@stu.sicau.edu.cn (Y.W.); 2021202089@stu.sicau.edu.cn (Y.Y.); 2021302105@stu.sicau.edu.cn (Y.Y.); 2021302109@stu.sicau.edu.cn (J.L.); 2021202092@stu.sicau.edu.cn (L.L.)

**Keywords:** salt stress, omostic regulation, antioxidant defence, metabolomics, white clover

## Abstract

White clover (*Trifolium repens*) is an excellent perennial cold-season ground-cover plant for municipal landscaping and urban greening. It is, therefore, widely distributed and utilized throughout the world. However, poor salt tolerance greatly limits its promotion and application. This study aims to investigate the difference in the mechanism of salt tolerance in relation to osmotic adjustment, enzymatic and nonenzymatic antioxidant defenses, and organic metabolites remodeling between salt-tolerant PI237292 (Trp004) and salt-sensitive Korla (KL). Results demonstrated that salt stress significantly induced chlorophyll loss, water imbalance, and accumulations of malondialdehyde (MDA), hydrogen peroxide (H_2_O_2_), and superoxide anion (O_2_^.−^), resulting in reduced cell membrane stability in two types of white clovers. However, Trp004 maintained significantly higher leaf relative water content and chlorophyll content as well as lower osmotic potential and oxidative damage, compared with KL under salt stress. Although Trp004 exhibited significantly lower activities of superoxide dismutase, peroxidase, catalase, ascorbate peroxidase, monodehydroasorbate reductase, dehydroascorbate reductase, and glutathione reductase than KL in response to salt stress, significantly higher ascorbic acid (ASA), dehydroascorbic acid (DHA), glutathione (GSH), glutathione disulfide (GSSG), ASA/DHA, and GSH/GSSG were detected in Trp004. These findings indicated a trade-off relationship between antioxidant enzymes and nonenzymatic antioxidants in different white clover genotypes adapting to salt stress. In addition, Trp004 accumulated more organic acids (glycolic acid, succinic acid, fumaric acid, malic acid, linolenic acid, and cis-sinapic acid), amino acids (serine, l-allothreonine, and 4-aminobutyric acid), sugars (tagatose, fructose, glucoheptose, cellobiose, and melezitose), and other metabolites (*myo*-inositol, arabitol, galactinol, cellobiotol, and stigmasterol) than KL when they suffered from the same salt concentration and duration of stress. These organic metabolites helped to maintain osmotic adjustment, energy supply, reactive oxygen species homeostasis, and cellular metabolic homeostasis with regard to salt stress. Trp004 can be used as a potential resource for cultivating in salinized soils.

## 1. Introduction

Excess soil salinity is an environmental pollution and a threat to the ecosystem and agriculture, and it causes an obvious decrease in the sustainability of crop production all around the world [1]. The area of salt-affected soils of about 8.3–11.7 Mkm^2^ has accounted for approximately 6% of the world’s total lands and 20% of the world’s irrigated agricultural lands, and it crosses all climate zones and mainlands [2]. High salinity results in damage to plant development and reduction in crop yield by two main primary stresses, “osmotic stress and ionic stress” [3]. On the one hand, plants suffer from ion toxicity and nutrient imbalance because they are compelled to absorb redundant salty ions after being subjected to salt stress [4]. On the other hand, low rhizospheric water potential induced by soil salinity prevents roots from taking in water, leading to physiological drought [5]. In order to maintain osmotic equilibrium, plants usually build up multiple osmolytes such as carbohydrates, amino acids, organic acids, secondary metabolites, etc. [6]. An earlier study has shown that exogenous chitosan-induced accumulations of γ-aminobutyric acid (GABA) and sucrose contribute to better osmotic balance and energy metabolism in creeping bentgrass (*Agrostis stolonifera*) during salt stress [7]. Enhanced accumulation and metabolism of various amino acids such as GABA, glutamic acid, alanine, proline, and cysteine were beneficial to water and metabolic homeostases in the leaves of creeping bentgrass under salt stress [8,9]. In addition, the exogenous application of secondary metabolite triacontanol promoted the concentration of carbohydrates, which improves the salt tolerance of pomegranate (*Punica granatum*) plants [10]. The study of Cheng et al. (2024) also found that the spermidine or spermine pretreatment improved the salt tolerance of white clover (*Trifolium repens*) by enhancing the accumulation of diverse amino acids, organic acids, sugars, and other metabolites [11]. These previous researchers highlight the positive roles of organic metabolites in osmotic adjustment (OA), osmprotection, and energy metabolism when plants undergo salt stress.

Salt-induced osmotic and ionic stresses further cause a series of secondary stresses, particularly oxidative stress, as a consequence of a burst of reactive oxygen species (ROS) in the plants [12]. The ROS at a low level is considered as essential signaling molecules for plant growth, development, and stress adaptation in the early stages of stress response, but the persistent and excess accumulation of ROS is toxic to membrane systems, organelle, and all other active substrates in plant cells [13]. Plants have evolved various adaptive mechanisms for scavenging extra ROS to adapt to salt stress [14]. Among them, multiple antioxidant enzymes are of primary importance for minimizing cellular oxidative damage as they limit ROS concentration in plants after being subjected to salt stress such as superoxide dismutase (SOD), peroxidase (POD), catalase (CAT), and the enzymes involved in ascorbate–glutathione (ASA-GSH) cycle, including ascorbate peroxidase (APX), glutathione reductase (GR), dehydroascorbate reductase (DHAR), and monodehydroasorbate reductase (MDHR) [6]. Furthermore, a variety of nonenzymic antioxidants, such as ASA, GSH, *myo*-inositol, and stigmasterol, also help to reduce the accumulation of ROS, based on the oxidation–reduction reaction of these antioxidants or through directly cleaning up ROS [14,15,16,17]. It has been reported that activations of SOD, POD, CAT, APX, GR, etc., could be beneficial to better adaptation to high salt environments by different plant species [18,19,20]. The *myo*-inositol was an important nonenzymic antioxidant for ROS scavenging in white clover under salt stress [11,21]. In addition, halophyte Tamarix ramosissima significantly up-regulated key genes for the biosynthesis of *myo*-inositol and the accumulation of *myo*-inositol during salt stress, indicating *myo*-inositol could act as an osmotic regulator and ROS scavenger to maintain osmotic and redox equilibria [22].

The vast majority of crops, forages, and turfgrasses could be classified into two main categories: halophytes and glycophytes. Halophytic species have evolved their morphological, physiological, and anatomical adaptability to saline environments, whereas most glycophytic species are unable to complete the life cycle when they grow in high salt-affected soils [23]. However, there is a significant variation in salt tolerance among different glycophytic plants. For example, cotton (*Gossypium hirsutum*) is considered as a moderately salt-tolerant crop, and onion (*Allium cepa*) has been found to be more sensitive to salt stress than soybeans (*Glycine max*) [24,25,26]. Even within the same species, the significant difference in salt tolerance is ubiquitous among different genotypes. A total of 552 sunflower (*Helianthus annuus*) germplasm resources with different genetic backgrounds exhibited significant variations in salt tolerance, and more than 30 genotypes were identified as salt-tolerant materials [27]. Abdelrady et al. (2024) obtained new salt-tolerant barley lines with a better capacity for ion homeostasis, osmoprotection, and antioxidant defense, compared with ordinary barley (*Hordeum vulgare*) cultivars [28]. White clover is widely used as a leguminous ornamental grass and landscaping ground cover plant for environmental greening and ecological restoration around the world [29,30]. In our previous study, differential salt tolerance was detected among 37 white clover genotypes, which were collected from different ecoregions worldwide, and PI237292 (Trp004) or Korla (KL) was identified as the salt-tolerant or salt-sensitive genotype, respectively. In addition, better salt tolerance of Trp004 was related to the maintenance of lower sodium ion (Na^+^) content and higher potassium ion (K^+^) content in shoots for a more stable ion homeostasis when under salt stress [31]. However, the underlying mechanism of salt tolerance between these two genotypes still needs to be further studied. The objective of current study was to explore differential salt tolerance between Trp004 and KL, which was associated with changes in oxidative damage, enzymic and nonenzymic antioxidant systems, and global organic metabolites for osmotic and metabolic homeostasis under salt stress.

## 2. Results

### 2.1. Differences in Chlorophyll Content and Water Status Between Trp004 and KL Under Optimal Conditions and Salt Stress

Salt stress significantly restricted plant growth, and the inhibitory effect was more pronounced in KL than that in Trp004, as shown by phonotypic changes in Figure 1A. Chl a content was not significantly altered in the leaves of Trp004 but significantly decreased in leaves of KL in response to salt stress (Figure 1B). Salt stress induced significant declines in Chl b and total Chl contents in the leaves of two genotypes (Figure 1B). The ratio of Chl a to Chl b in two genotypes was not affected significantly by salt stress (Figure 1B). Trp004 maintained a 118%, 194%, 51%, and 85% higher ratio of Chl a to b, total Chl content, Chl a content, and Chl b content than Trp004 in leaves under salt stress, respectively (Figure 1B). No genotypic variation in leaf RWC was observed between the two materials under optimal conditions (Figure 1C). Leaf RWC of Trp004 and KL were significantly reduced by salt stress, but Trp004 has 26% higher RWC than KL under salt stress (Figure 1C). OP was not significantly different between Trp004 and KL under optimal conditions (Figure 1D). Salt stress induced a significant decline in OP in both genotypes, and the OP in leaves of Trp004 was 49.63% lower than KL under salt stress (Figure 1D).

### 2.2. Differences in Oxidative Damage and Antioxidant Metabolism Between Trp004 and KL Under Optimal Conditions and Salt Stress

Under optimal conditions, there were no differences in EL, MDA, H_2_O_2_, and O_2_^.−^ between KL and Trp004 (Figure 2A–D). As compared with KL+C, EL, MDA, H_2_O_2_, and O_2_^.−^ increased significantly in KL+S (Figure 2A–D). However, although Trp004+S exhibited significantly higher EL and O_2_^.−^ than Trp004+C, MDA and H_2_O_2_ contents of Trp004 were not significantly affected by salt stress (Figure 2A–D). KL+S maintained a 16%, 200%, 78%, and 42% higher EL, MDA, H_2_O_2_, and O_2_^.−^ than Trp004+S, respectively (Figure 2A–D).

Under salt stress, the activities of SOD, POD, and CAT increased to extremely high levels in KL. However, salt stress did not significantly change the activities of SOD and CAT in Trp004 but reduced the activity of POD significantly in Trp004 (Figure 3A–C). KL exhibited 259%, 686%, and 223% higher activities of SOD, POD, and CAT than KL under salt stress, respectively (Figure 3A–C). Salt stress significantly improved activities of APX, MDHR, DHAR, and GR in KL as well as GR activity in Trp004 but had no significant effect on activities of APX, MDHAR, and DHAR in Trp004 (Figure 4A–D). KL maintained 2134%, 127%, 209%, and 114% increases in activities of APX, MDHR, DHAR, and GR than Trp0004 under salt stress, respectively (Figure 4A–D). Significant accumulations of ASA and GSH were induced by salt stress in Trp004 but not in KL (Figure 5A,D). Salt stress significantly inhibited the accumulation of DHA in both Trp004 and KL and also inhibited the accumulation of GSSG in Trp004 but did not alter the accumulation of GSSG in KL (Figure 5B,E). The ratio of ASA to GSSG significantly increased in both genotypes in response to salt stress, but the ratio of GSH to GSSG only significantly increased in Trp004 under salt stress (Figure 5C,F). The ASA/DHA and GSH/GSSG in Trp004 were 2.3 times and 11.0 times higher than KL under salt stress, respectively (Figure 5C,F).

### 2.3. Differences in Metabolites Profile Between Trp004 and KL Under Optimal Condition and Salt Stress

A total of sixty-eight differentially expressed metabolites (DEMs) were identified and quantified, including twenty-nine organic acids, seven amino acids, eighteen sugars, and fourteen other metabolites (Figure 6A). The heat map showed overall changes in the contents of 68 DEMs based on four different comparison groups (KL+S vs. KL+C, Trp004+S vs. Trp004+C, Trp004+C vs. KL+C, and Trp004+S vs. KL+S) (Figure 6A). In KL+S vs. KL+C, Trp004+S vs. Trp004+C, Trp004+C vs. KL+C, and Trp004+S vs. KL+S, the 63%, 63%, 22%, and 22% metabolites were down-regulated, and the 16%, 24%, 24% and 37% metabolites were significantly up-regulated, and the rest of metabolites were not significant difference between two treatments (Figure 6B). Salt stress resulted in a significant decrease in the content of organic acids in two materials, as shown by KL+S vs. KL+C and Trp004+S vs. Trp004+C (Figure 6C). The content of amino acids in Trp004+S was significantly higher than that in KL+S (Trp004+S vs. KL+S). Salt stress induced a significant decline in the content of amino acids in KL (KL+S vs. KL+C) and also decreased the content of sugars in Trp004 (Trp004+S vs. Trp004+C). In addition, salt stress decreased the content of other metabolites in KL (KL+S vs. KL+C) but increased the content of other metabolites in Trp004 (Trp004+S vs. Trp004+C). The content of other metabolites in Trp004 was 181% higher than KL under salt stress (Trp004+S vs. KL+S) (Figure 6C).

Under optimal conditions, the contents of glycolic acid, linolenic acid, and cis-sinapinic acid in the leaves of Trp004 were significantly higher than those in the leaves of KL (Trp004+C vs. KL+C) (Figure 7A). Under stress conditions, the contents of glycolic acid, succinic acid, fumaric acid, malic acid, linolenic acid, and cis-sinapinic acid in the leaves of Trp004 were significantly higher than those in the leaves of KL (Trp004+S vs. KL+S) (Figure 7A). Salt stress significantly reduced the contents of serine, allothreonine, and GABA in KL (KL+S vs. KL+C) but not in Trp004 (Trp004+S vs. Trp004+C) (Figure 7B). Trp004 exhibited significantly higher contents of serine, allothreonine, and GABA than KL under salt stress, as demonstrated by Trp004+S vs. KL+S (Figure 7B). Tagatose, fructose, cellobiose, and melezitose in two genotypes significantly reduced in response to salt stress (Trp004+S vs. Trp004+C and KL+S vs. KL+C), whereas the accumulation of glucoheptose significantly increased in Trp004 under salt stress (Trp004+S vs. Trp004+C) (Figure 7C). Trp004+S exhibited significantly higher tagatose, fructose, glucoheptose, cellobiose, and melezitose than KL+S (Trp004+S vs. KL+S) (Figure 7C). Significantly higher contents of arabitol and cellobiotol were detected in Trp004 compared with KL under optimal conditions (Trp004+C vs. KL+C) (Figure 7D). The accumulations of *myo*-inositol, arabitol, galactinol, cellobiotol, and stigmasterol in Trp004 were significantly higher than that in KL under salt stress (Trp004+S vs. KL+S) (Figure 7D). Figure 8 shows integrative metabolic pathways of antioxidant defense and the metabolism of various organic metabolites involved in glucose metabolism, tricarboxylic acid (TCA) cycle, GABA shunt, etc.

## 3. Discussion

Leaf wilting and chlorisis are typical symptoms of high salt damage due to salt-induced primary stresses such as osmotic stress and ion toxicity, as well as secondary stresses such as oxidative damage [32,33]. Additionally, plants exhibit dwarf phenotype and low biomass in response to salt stress [34,35]. Photosynthetic pigments play an important role in converting light energy into chemical energy for plant growth and development [36]. As salt-sensitive organelles, Chl loss and impaired chloroplasts led to a slowdown in plant growth under salt stress [37]. Previous studies have demonstrated that Trp004 maintained better growth associated with higher Chl content and net photosynthetic rate than KL under salt stress [31], which is consistent with the current study. This is similar to the finding of Jawaria et al., who found that the salt-tolerant eggplant (*Solanum melongena*) variety ICS-BR-1351 exhibited a smaller decrease in Chl content and the best growth than other varieties in response to salt stress [38]. Analysis of metabolomics demonstrated that salt stress significantly reduced accumulations of intermediates of the TCA cycle, such as citric acid and fumaric acid in Trp004 and KL, as well as malic acid and succinic acid in KL, indicating inhibitory TCA cycle for energy metabolism. However, Trp004 exhibited significantly higher contents of all intermediates of the TCA cycle than KL under salt stress, which could be associated with better maintenance of energy production for Trp004 growth. It has been found that enhanced accumulations of citric acid, malic acid, and α-ketoglutarate for TCA cycle could be related to putrescine-promoted seed germination and seedling growth when white clover seeds germinated in salinity condition [21]. In addition, organic acids including succinic acid, fumaric acid, malic acid, glycolic acid, linolenic acid, and cis-sinapinic acid exhibit important roles in ionization balance and antioxidant defense as a result of their weak acidity and ionization abilities [39,40].

Salt stress-induced the burst of ROS in chloroplasts, mitochondria, peroxisomes, and extracellular vesicles directly oxidizes cell membrane systems and proteins, resulting in Chl degradation and destruction of chloroplast ultrastructure [41,42,43]. Therefore, the homeostasis between generation and elimination of ROS is of primary importance for plants to survive salt stress [44,45]. It has been found that salt-tolerant rice (*Oryza sativa*) cultivar Vyttila accumulated significantly lower contents of H_2_O_2_ and O_2_^.−^ than salt-sensitive Aiswarya after being subjected to the same salt concentration and duration of stress in virtue of better redox equilibrium in cells [46]. Acetic acid priming can also mitigate salt stress by promoting ROS scavenging in perennial ryegrass (*Lolium perenne*) [47]. In our current study, salt stress significantly induced accumulations of MDA, H_2_O_2_, and O_2_^.−^ in leaves of KL, resulting in a significant decline in cell membrane stability, as reflected by a significant increase in EL level. However, salt stress only significantly enhanced the accumulation of O_2_^.−^ and EL levels in Trp004 associated with lower cellular oxidative damage.

The critical role of enzymatic antioxidant systems in detoxifying ROS has been widely proved in various plant species, including white clover, under salt stress since SOD acts as a main scavenger of O_2_^.−^, and CAT and POD are responsible for scavenging H_2_O_2_ [7,48,49]. Another classic antioxidant pathway is the ASA-GSH cycle involved in nonenzymatic antioxidants AsA and GSH [50]. The ability of ROS scavenging in this defense system depends on the redox reaction between ASA and DHA or GSH and GSSH [51]. Interestingly, salt stress activated various antioxidant enzymes such as SOD, POD, CAT, MDHR, and DHAR in KL but not in Trp004, while nonenzymatic antioxidants, such as ASA and GSH, only significantly increased in Trp004 under salt stress. In addition, Trp004 also maintained significantly higher ASA/DHA and GSH/GSSG than KL in response to salt stress, indicating a better redox state. A recent study by Gao et al. found that salt stress significantly weakened enzymatic antioxidant systems in peanut (*Arachis hypogaea*) seedlings but improved the biosynthesis of nonenzymatic antioxidants, including ASA and flavonoids, to alleviate oxidative damage. However, exogenous application of calcium could effectively eliminate salt-induced accumulation of ROS in peanut seedlings by promoting the activities of antioxidant enzymes and inhibiting the accumulation of flavonoids, indicating a trade-off relationship between antioxidant enzyme activities and nonenzymatic metabolites [52]. Trp004 suffered from lower oxidative damage than KL, which could be related to lower salt-caused water stress and ionic toxicity, as reflected by higher leaf RWC in the current study, as well as the lower sodium content and higher ratio of potassium to sodium in our previous study [31]. On the other hand, lower primary stresses allowed Trp004 to mobilize nonenzymatic antioxidants to alleviate the ROS damage induced by high salinity.

The high osmotic pressure of saline soil solution seriously hinders water absorption and transport by plant roots [53]. Water shortage in plants not only induces ROS burst but also limits photosynthetic carbon assimilation, leading to metabolic deficit under salt stress [54,55,56]. Halophytes have a superior ability to compartmentalize inorganic salt ions such as Na^+^ into vacuoles for OA, but the ability in glycophytic plants is weak. Glycophytes generally rely on the synthesis of organic metabolites to reduce OP for water homeostasis [23]. As an important energy source and osmotic regulatory substance, free amino acids and sugar are crucial in plant growth and development [57,58]. The study of Noreen et al. has demonstrated that the exogenous application of *Moringa olifera* leaf extract significantly alleviated the decline in leaf RWC caused by salt stress by improving the accumulation of free amino acids in pea (*Pisum sativum*) leaves [59]. Biochar and melatonin-regulated water homeostasis in borage (*Borago officinalis*) plants associated with accumulations of proline and soluble carbohydrates contributed to reduced OP under salt stress [60]. Moreover, wheat (*Triticum aestivum*) cultivar Gimeza 9 with lower salt sensitivity could maintain significantly higher RWC and various amino acid contents (asparatic acid, serine, thereonine, glutamic acid, proline, glysine, alanine, valine, methionine, leucine, isoleucine, phenylalanine, tyrosine, histidine, lysine, arginine, arginine, and cysteine) than salt-sensitive Giza 168 throughout the entire period of salt stress [61]. Similar findings were shown in our current study, and Trp004 accumulated more multiple amino acids (serine, allothreonine, glutaconic acid, and GABA) and surges (tagatose, fructose, glucoheptose, cellobiose, and melezitose) compared with KL in response to salt stress. These organic osmolytes could help Trp004 to maintain better OA and water balance under salt stress. It is well-known that GABA functions as a regulator of abiotic stress defense by improving carbon and nitrogen metabolism, energy supply, OA, and ROS detoxification in plants [62]. Exogenous GABA-induced accumulations of endogenous amino acids (GABA, serine, alanine, glutamic acid, cysteine, etc.) and sugars (glucose, fructose, maltose, trehalose, galactose, etc.) were positively related to improved OA, water homeostasis, and metabolic balance in creeping bentgrass suffering salt stress [8]. Furthermore, no significant difference in transpiration rate between Trp004 and KL was detected under optimal and salinity conditions [31]. Together with the findings in the current study, the stronger ability of water regulation in Trp004 could mainly depend on the accumulation of organic metabolites. In addition to the beneficial roles of tagatose, fructose, cellobiose, melezitose, etc., in energy sources and OA, they also act as important signaling molecules to activate defensive response to multi-stresses [63,64,65,66,67].

Under salt stress, Trp004 also significantly accumulated more other metabolites, including *myo*-inositol, arabitol, galactinol, cellobiotol, and stigmasterol, as compared with KL. As known as a growth-regulating factor, *myo*-inositol modulates complex cellular metabolic pathways and stress responses [68]. Many studies have shown that *myo*-inositol could act as an osmotic regulator, ROS scavenger, and signaling molecule to enhance salt tolerance in plants [22,69,70]. Plant growth-promoting rhizobacteria conferred salt tolerance to tomato plants by up-regulating the accumulation of *myo*-inositol [71]. The exogenous supply of *myo*-inositol provoked accumulations of free amino acids, soluble sugars, and other metabolites for ROS homeostasis and osmotic balance in different plant species under salt stress [16,68,72]. Arabitol has been found in plants, fungi, and other living beings, fulfilling the function of osmoprotection or carbohydrate storage [73]. Cellobiotol and galactinol are presumed to be involved in cell wall carbohydrate metabolism [74]. Nishizawa et al. found out that high intracellular level of galactinol in transgenic *Arabidopsis thaliana* overexpressing a *galactinol synthase* was correlated with increased tolerance to methylviologen treatment, salinity stress, or chilling stress owing to the role of galactinol in OA and osmoprotection [75]. In addition, stigmasterol is a common plant sterol that not only helps to maintain cell membrane stability but also plays a critical role in the adaptation to stress as an antimicrobial and antioxidant [76]. For example, exogenous stigmasterol increased the salt tolerance of faba bean (*Vicia faba*) plants by enhancing antioxidant systems [77]. The study of Bassuany et al. also found that the application of stigmasterol significantly increased GSH and carbohydrates in flax (*Linum usitatissimum*) plants to decrease oxidative damage and growth retardant induced by salt stress [78]. In the current study, salt stress significantly promoted the contents of *myo*-inositol, galactinol, and stigmasterol only in Trp004 but not in KL. This indicated that accumulations of these metabolites in Trp004 could play positive roles in OA, energy supply, ROS scavengers, and metabolic regulation under salt stress, thereby improving defensive response and better growth.

## 4. Materials and Methods

### 4.1. Plant Materials and Treatments

Two white clover genotypes, salt-tolerant Trp004 and salt-sensitive KL, were utilized as the materials. Trp004 is provided by NPGS, and KL is a common commercial cultivar [31]. For vegetative propagation, four stem segments (2 stem nodes per stem segment) were planted in each PVC tube (12 cm in diameter and 20 cm high), which were filled with sands and nutrient soils (1:1, v:v). All plants were watered with 200 mL of 1/2 Hoagland’s solution twice a week [79]. After being cultivated in growth chambers (23/19 °C (day/night), 60% relative humidity, and 700 μmol·m^−2^·s^−1^ PAR) for 40 days, plants were subjected to salt stress. Four treatments were set as follows: (1) Trp004+C, Trp004 grew under optimal condition; (2) KL+C, KL grew under optimal condition; (3) Trp004+S, Trp004 grew under salt stress; (4) KL+S, KL grew under salt stress. Each treatment included four independent replications (four pots). For salt stress, plants in each pot were irrigated by using 200 mL of 150 mM NaCl on the first day, 200 mL of 200 mM NaCl on the fourth day, and 200 mL of 250 mM NaCl on the eighth day. Leaf samples were collected for determination of physiological parameters, antioxidant enzyme activities, and organic metabolites on 16th day of normal cultivation and salt stress.

### 4.2. Determination of Chlorophyll Content and Water Status

For the determination of chlorophyll (Chl) content, 0.1 g of leaf sample was soaked completely in 10 mL of dimethyl sulfoxide (DMSO) for 48 h in the dark, and the absorbancy of 200 μL of extract solution was detected at 645 nm and 663 nm by using UV-2102PC UV spectrophotometer [80]. For determination of relative water content (RWC), fresh leaves were collected and weighed to record fresh weight. These leaves were wrapped with absorbent paper and then put into a 50 mL centrifuge tube containing 25 mL of deionized water for 24 h. After being taken out from water, the moisture on leaf surface was wiped gently, and saturated fresh weight was recorded. These leaves were then placed in an oven at 105 °C for 3 h and 70 °C for two days. Dry weight was recorded. The formula RWC (%) = (fresh weight-dry weight)/(saturated fresh weight-dry weight) × 100% was used to calculate leaf RWC [81]. For the osmotic potential (OP) of leaf tissue, 0.3 g of fresh leaves were wrapped with absorbent paper and then put into 50 mL of deionized water at 4 °C for 24 h. After the moisture on leaf surface was wiped off, these leaves were put into a 1.5 mL centrifuge tube, which was then immerged in liquid nitrogen for 10 min. After being thawed, the leaf sap was squeezed out from the leaves, and the osmolality of 10 μL sap was detected by using a WESCOR-5600 osmometer. The OP was calculated based on Mpa = −osmolality × 2.58 × 10^−3^ [82].

### 4.3. Determination of Cell Membrane Stability, Oxidative Damage, and Antioxidant Metabolism

For electrolyte leakage (EL), the initial conductivity (S1) of the solution was measured using a DDS-11A conductivity meter after 0.1 g of fresh leaves were completely immersed in 25 mL distilled water for 24 h. Leaves were boiled for 30 min and then cooled down to room temperature. The final conductivity of solution (S2) was measured. The EL can be calculated using the formula EL = (S1/S2) × 100% [83]. To detect malondialdehyde (MDA) content and antioxidant enzyme activities, about 0.1 g leaf tissues were ground into the fine powder in liquid nitrogen and mixed well with 1.5 mL of 150 mM phosphate buffer (pH 7.0) and then centrifuged at 12,000× *g* for 15 min to obtain supernatant. The kinetic curves of POD, CAT, APX, GR, DHAR, and MDHR activities were measured every 10 s at 470, 240, 290, 340, 265, and 340 nm, respectively [84,85,86]. The SOD activity was determined by nitrogen blue tetrazole (NBT) method at 560 nm [87]. Assay methods have been clearly presented in detail in our previous study [88]. For the determination of MDA, 1 mL of reaction solution (20% trichloroacetic acid and 0.5% thiobarbituric acid) was mixed with 0.5 mL of supernatant and then incubated in 95 °C water bath for 30 min. After being cooled quickly to room temperature on the ice and centrifuged at 10,000× *g* for 10 min, the absorbance of the supernatant was measured at 532 and 600 nm [89]. H_2_O_2_ and O_2_^.−^ were determined according to the methods of [90] and [91], respectively. Assay methods of H_2_O_2_ and O_2_^.−^ contents have been clearly presented in detail in our previous study [92]. Suzhou Comin Biotechnolgy Co., Ltd. (Suzhou, China) provided the kits for determinations of ASA (ASA-2A-W), dehydroascorbic acid (DHA) (DHA-1-W), glutathione disulfide (GSSG) (GSSG-2-W), and GSH (GSH-1-W) contents.

### 4.4. Metabolomics Analysis

A lyophilizer (LGJ-10C) was used to freeze-dry fresh leaves which were ground into fine powder after freeze-drying. A total of 20 mg fine powder was used to extract organic metabolites according to the previous method [92]. The analysis and identification of metabolites depended on the full two-dimensional gas chromatography/time of flight mass spectrometry (GC-TOFMS, Pegasus 4D, LECO Corporation, St. Joseph, MI, USA), commercial compound libraries NIST 2005 (PerkinElmer Inc., Waltham, MA, USA), and Feihn metabolites (Feihnlab., Davis, CA, USA).

### 4.5. Statistical Analysis

Microsoft Excel 2013 (Microsoft, Redmond, WA, USA) was used for statistics and preliminary processing of the original data. Significant differences (*p* ≤ 0.05) were detected by using SPSS 26.0 (IBM, Armonk, NY, USA) based on two-way analysis of variance (ANOVA) together with the Tukey’s test, and results of two-way ANOVA are demonstrated in Appendix A. All data were demonstrated with mean ± standard deviation (mean ± SD). Charts were drawn by using Origin (OriginLab, Northampton, MA, USA) and TBtools (Guangzhou, China).

## 5. Conclusions

Trp004 had significantly better salt tolerance than KL, which was associated with additional accumulation of nonenzymatic antioxidants (ASA and GSH) instead of the activating of various antioxidant enzyme activities (SOD, CAT, POD, APX, GR, DHAR, and MDHAR). These findings indicated a trade-off relationship between antioxidant enzyme activities and nonenzymatic antioxidants in different white clover genotypes adapting to salt stress. Compared with KL, Trp004 also accumulated more organic acids (glycolic acid, succinic acid, fumaric acid, malic acid, linolenic acid, and cis-sinapic acid) for energy metabolism, ionization balance, and antioxidant defense; amino acids and sugars (serine, l-allothreonine, 4-aminobutyric acid, tagatose, fructose, glucoheptose, cellobiose, and melezitose) for energy supply, OA, and ROS detoxification; and other metabolites (*myo*-inositol, arabitol, galactinol, cellobiotol, and stigmasterol) involved in OA and ROS homeostasis. This occurs when they suffered from the same salt concentration and duration of salt stress. Further study will focus on the differential salt tolerance of the two genotypes in roots in the future.

## Figures and Tables

**Figure 1 plants-14-00145-f001:**
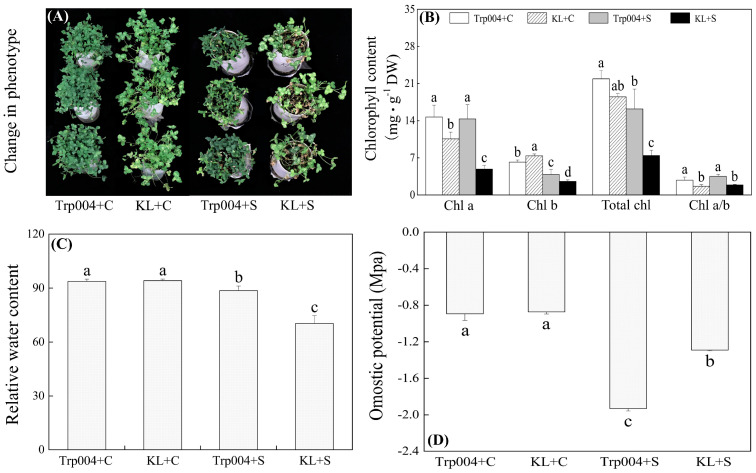
Differences in (**A**) phenotypic change, (**B**) chlorophyll content, (**C**) relative water content (RWC), and (**D**) omostic potential (OP) in leaves of Trp004 and KL under salt stress and optimal conditions. Different letters indicate significant differences among four groups (*p* ≤ 0.05).

**Figure 2 plants-14-00145-f002:**
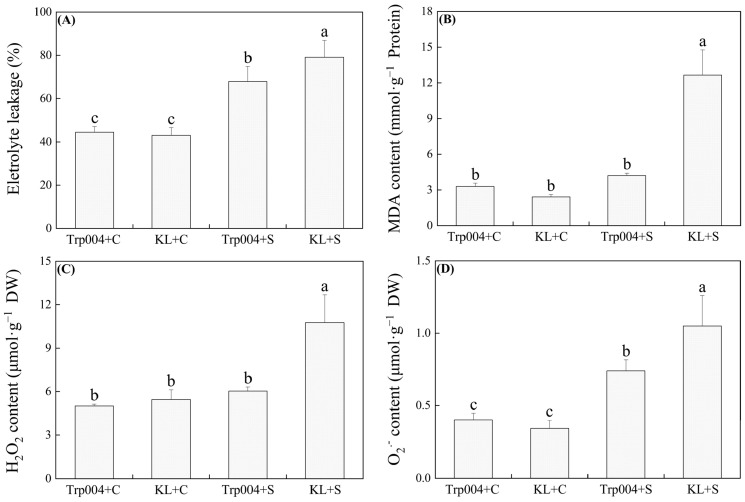
Differences in (**A**) electrolyte leakage (EL), (**B**) malondialdehyde (MDA), (**C**) hydrogen peroxide (H_2_O_2_), and (**D**) superoxide anion (O_2_^.−^) content in leaves of Trp004 and KL under salt stress and optimal conditions. Different letters indicate significant differences among four groups (*p* ≤ 0.05).

**Figure 3 plants-14-00145-f003:**
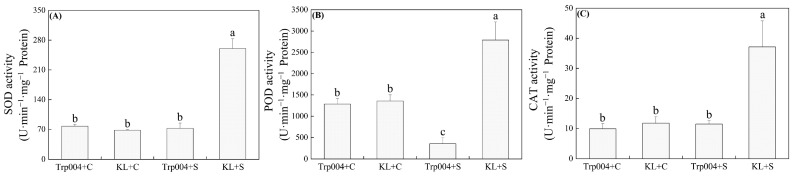
Differences in activities of (**A**) superoxide dismutase (SOD), (**B**) peroxidase (POD), and (**C**) catalase (CAT) in leaves of Trp004 and KL under salt stress and optimal conditions. Different letters indicate significant differences among four groups (*p* ≤ 0.05).

**Figure 4 plants-14-00145-f004:**
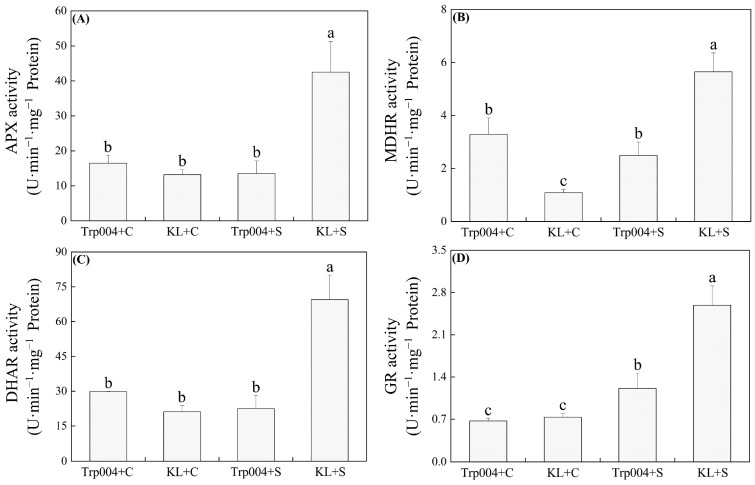
Differences in activities of (**A**) ascorbate peroxidase (APX), (**B**) monodehydroasorbate reductase (MDHR), (**C**) dehydroascorbate reductase (DHAR), and (**D**) glutathione reductase (GR) in leaves of Trp004 and KL under salt stress and optimal conditions. Different letters indicate significant differences among four groups (*p* ≤ 0.05).

**Figure 5 plants-14-00145-f005:**
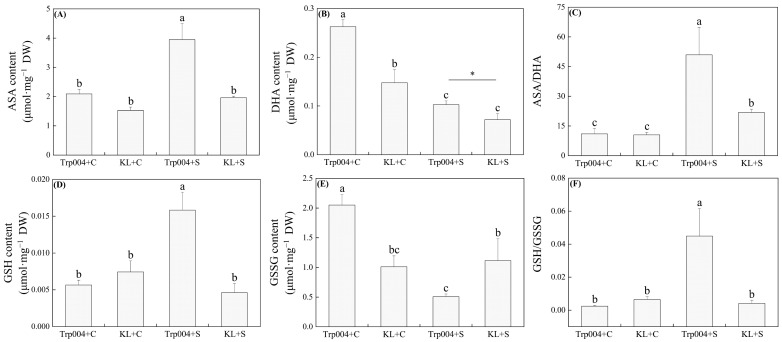
Differences in contents of (**A**) ascorbic acid (ASA), (**B**) dehydroascorbic acid (DHA), (**C**) the ratio of ASA to DHA, (**D**) glutathione (GSH), (**E**) glutathione disulfide (GSSG), and (**F**) the ratio of GSH to GSSG in leaves of Trp004 and KL under salt stress and optimal conditions. Different letters indicate significant differences among the four groups, and the “*” represents a significant difference between the two treatments (*p* ≤ 0.05).

**Figure 6 plants-14-00145-f006:**
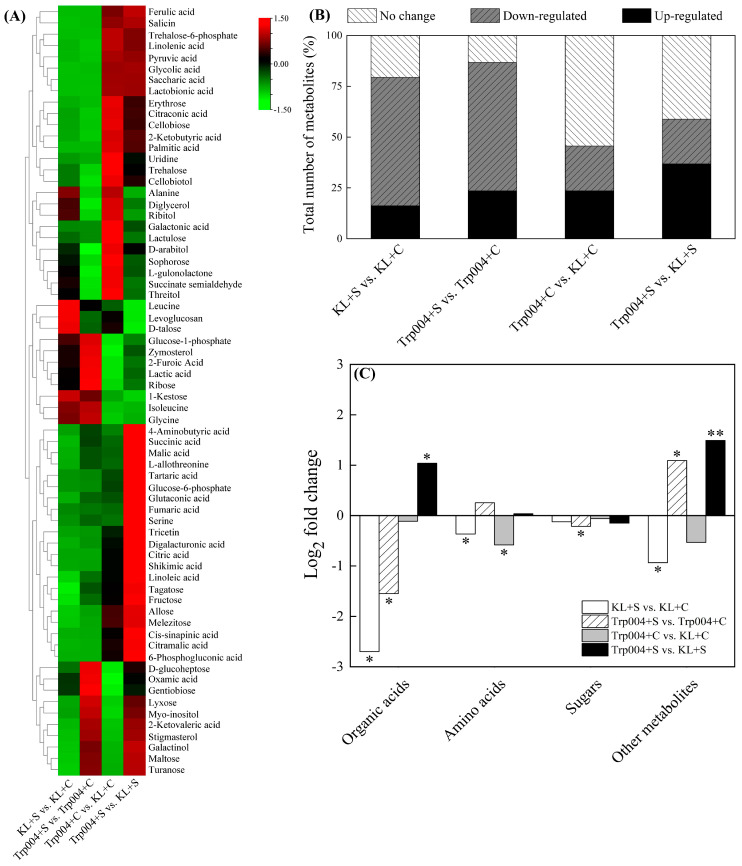
Changes in (**A**) heat map of 68 metabolites, (**B**) percentage of total number of metabolites in each group, and (**C**) total content of organic acids, amino acids, sugars, and other metabolites in four comparison groups (KL+S vs. KL+C, Trp004+S vs. Trp004+C, Trp004+C vs. KL+C, and Trp004+S vs. KL+S). Log_2_ (fold change) ratios were demonstrated in heat map. The “*” or “**” represents a significant difference between two treatments at *p* ≤ 0.05 or *p* ≤ 0.01, respectively.

**Figure 7 plants-14-00145-f007:**
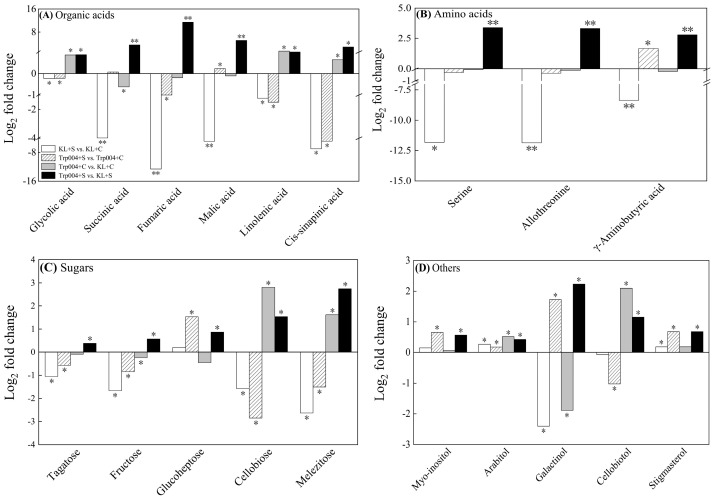
Differences in contents of (**A**) organic acids, (**B**) amino acids, (**C**) sugars, and (**D**) other metabolites in leaves of Trp004 and KL under optimal conditions and salt stress. The “*” or “**” represents a significant difference between two treatments at *p* ≤ 0.05 or *p* ≤ 0.01, respectively.

**Figure 8 plants-14-00145-f008:**
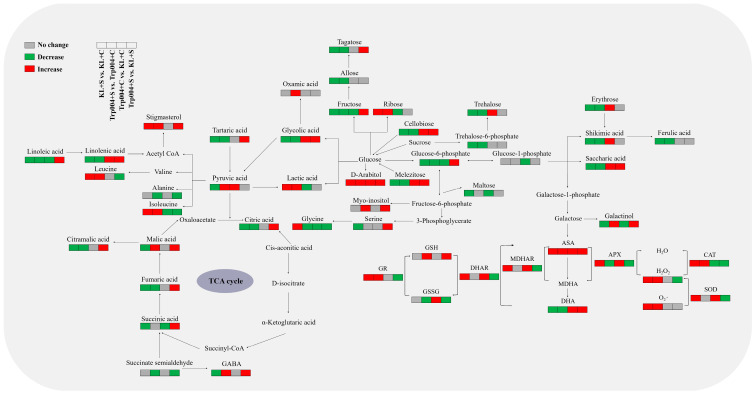
A metabolic pathway map is involved in 38 different metabolites. Different colors indicate significant difference between specific two treatments (KL+S vs. KL+C, Trp004+S vs. Trp004+C, Trp004+C vs. KL+C, or Trp004+S vs. KL+S). Red, green, or grey represent a significant increase, a significant decrease, or no change, respectively.

## Data Availability

All of the datasets supporting the results of this article are included within the article.

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
