# Peer review of "Trade-Off Between Enzymatic Antioxidant Defense and Accumulation of Organic Metabolite Affects Salt Tolerance of White Clover Associated with Redox, Water, and Metabolic Homeostases"

_plants, 2025, doi:10.3390/plants14020145_

Round 1

Reviewer 1 Report

Comments and Suggestions for Authors

- The manuscript is poorly formatted (superscripts and subscripts)

- What was the basis behind the chosen NaCl concentrations? 200 mM NaCl seems an exaggerated concentration. Is it relevant for the environment where the species grows?

- Why not analyse the roots as they are the first organ in contact with salt and might be heavily involved in the plants' tolerance?

- If the authors did a two-way ANOVA then please provide the ANOVA results (i.e. p-value for each factor and their interaction) because the Tukey comparing all groups amongst each other might be flawed depending on the ANOVA result.

- The results of your previous study should be better detailed on the introduction (i.e. what are the already known mechanisms mediating the salt tolerance, for instance lower accumulatio of Na, or lower translocation to shoots)

Author Response

  1. The manuscript is poorly formatted (superscripts and subscripts)

Response: Thank you very much for professional review and giving us some good suggestions to improve our study. We have revised the manuscript carefully according to all suggestions. Format errors have been revised throughout the whole manuscript.

  1. What was the basis behind the chosen NaCl concentrations? 200 mM NaCl seems an exaggerated concentration. Is it relevant for the environment where the species grows?

Response: Thanks for your insightful comments. In this experiment, different concentrations of NaCl solutions were irrigated in soils to induce salt stress (line 372-376): plants in each pot were irrigated by using 200 ml of 150 mM NaCl on the first day, 200 ml of 200 mM NaCl on the fourth day, and 200 ml of 250 mM NaCl on the eighth day. This processing method allows plants to adapt to the salt stress without dying immediately. Subsequently, we chose 250 mM concentration based on the following reasons:

  • In the preliminary tests, final concentration of 250 mM NaCl triggered obvious physiological difference between two white clover genotypes.
  • In previous study,final concentration of 250 mM NaCl has been identified as a effective dose to screen different white clover genotypes with different salt tolerance (Li et al., 2022).

Li, Z., Geng, W., Tan, M., Ling, Y., Zhang, Y., Zhang, L., & Peng, Y. (2022). Differential responses to salt stress in four white clover genotypes associated with root growth, endogenous polyamines metabolism, and sodium/potassium accumulation and transport. Frontiers in Plant Science, 13, 896436.

  • In nature, the salt concentrationin saline-alkali land areas can even exceed 1.6% (274 mM).
  1. Why not analyse the roots as they are the first organ in contact with salt and might be heavily involved in the plants' tolerance?

Response: Thanks you so much for good suggestion. Yes, we totally agree with you. Roots are the first organ in contact with salt and heavily involved in the plants' tolerance to salt stress. This is what we will do in our future study. Current study has provided multiple data including physiological and metabolic parameters to explore differential response between two white clover genotypes. We will further reveal different mechanism in roots in our future study. Relevant explanation has been added in the section of Conclusions: Further study will focus on differential salt tolerance of two genotypes in roots in future (line 453-454).

  1. If the authors did a two-way ANOVA then please provide the ANOVA results (i.e. p-value for each factor and their interaction) because the Tukey comparing all groups amongst each other might be flawed depending on the ANOVA result.

Response: Thanks. Yes, significant differences (P≤0.05) were detected based on two-way analysis of variance (ANOVA) together with the Tukey test. The two-way ANOVA results have been provided as a supplementary table S1.

  1. The results of your previous study should be better detailed on the introduction (i.e. what are the already known mechanisms mediating the salt tolerance, for instance lower accumulation of Na, or lower translocation to shoots)

Response: Thanks for wise suggestion, and we added the already known mechanisms mediating the salt tolerance in previous study (line 108-110).

Reviewer 2 Report

Comments and Suggestions for Authors

Comments to the manuscript entitled The tradeoff between enzymatic antioxidant defense and accumulation of organic metabolite affects salt tolerance of white clover associated with redox, water, and metabolic homeostases” (Manuscript Number: plants-3364818) written by Min Zhou, Yuting Wu, Yuchen Yang, Yan Yuan, Junnan Lin, Long Lin, Zhou Li.

General comment

In reviewed manuscript authors investigated effect of salt stress, generated by NaCl, on two genotypes of white clover growth and physiological response. Presented research is interesting. Authors assessed, a lot of different physiological parameters, important in evaluation of plant response to salt stress. There are no doubts that experiments were executed carefully and manuscript is well written. However authors did not avoid some minor errors.

Material and methods, chapter 4.3 (Determination of cell membrane stability, oxidative damage, and antioxidant metabolism) must be written in most details, especially part concerning evaluation of enzymes’ activities and non-enzymatic antioxidants’ content.

Conclusion, must be rewritten to be less of result summary and more explain what type of mechanisms of salt tolerance are observed in tested plants.

Author Response

In reviewed manuscript authors investigated effect of salt stress, generated by NaCl, on two genotypes of white clover growth and physiological response. Presented research is interesting. Authors assessed, a lot of different physiological parameters, important in evaluation of plant response to salt stress. There are no doubts that experiments were executed carefully and manuscript is well written. However authors did not avoid some minor errors.

Response: Thank you very much for professional review and giving us some good suggestions to improve our study. We have revised the manuscript carefully according to all suggestions.

  1. Material and methods, chapter 4.3 (Determination of cell membrane stability, oxidative damage, and antioxidant metabolism) must be written in most details, especially part concerning evaluation of enzymes’activities and non-enzymatic antioxidant’ 

Response: Thank you very much for careful review. In order to avoid a high similarity rate in the paper during the plagiarism check, assay methods of some conventional parameters such as cell membrane stability, oxidative damage, and antioxidant enzymes were simplified. However, these assay methods in details have been clearly presented in our previous studies. We have added relevant explanations in revised manuscript (line 406-414). 

  1. Conclusion, must be rewritten to be less of result summary and more explain what type of mechanisms of salt tolerance are observed in tested plants.

Response: Thank you for good suggestion. The conclusion has been rewritten to be less of result summary and more explanation of potential mechanisms in revised manuscript (line 434-456).